# Optimization of SNAP-25 and VAMP-2 Cleavage by Botulinum Neurotoxin Serotypes A–F Employing Taguchi Design-of-Experiments

**DOI:** 10.3390/toxins11100588

**Published:** 2019-10-11

**Authors:** Laura von Berg, Daniel Stern, Jasmin Weisemann, Andreas Rummel, Martin Bernhard Dorner, Brigitte Gertrud Dorner

**Affiliations:** 1Biological Toxins, Centre for Biological Threats and Special Pathogens, Robert Koch Institute, Seestr. 10, 13353 Berlin, Germany; vonberg.laura@gmail.com (L.v.B.); sternd@rki.de (D.S.); dornerm@rki.de (M.B.D.); 2Institut für Toxikologie, Medizinische Hochschule Hannover, Carl-Neuberg-Str. 1, 30625 Hannover, Germany; weisemann.jasmin@mh-hannover.de (J.W.); rummel.andreas@mh-hannover.de (A.R.)

**Keywords:** botulinum neurotoxins, detection, substrate cleavage, taguchi design-of-experiments, SNARE

## Abstract

The detection of catalytically active botulinum neurotoxins (BoNTs) can be achieved by monitoring the enzymatic cleavage of soluble NSF (N-ethylmaleimide-sensitive-factor) attachment protein receptor (SNARE) proteins by the toxins’ light chains (LC) in cleavage-based assays. Thus, for sensitive BoNT detection, optimal cleavage conditions for the clinically relevant A–F serotypes are required. Until now, a systematic evaluation of cleavage conditions for the different BoNT serotypes is still lacking. To address this issue, we optimized cleavage conditions for BoNT/A–F using the Taguchi design-of-experiments (DoE) method. To this aim, we analyzed the influence of buffer composition (pH, Zn^2+^, DTT (dithiothreitol), NaCl) as well as frequently used additives (BSA (bovine serum albumin), Tween 20, trimethylamine N-oxide (TMAO)) on BoNT substrate cleavage. We identified major critical factors (DTT, Zn^2+^, TMAO) and were able to increase the catalytic efficiency of BoNT/B, C, E, and F when compared to previously described buffers. Moreover, we designed a single consensus buffer for the optimal cleavage of all tested serotypes. Our optimized buffers are instrumental to increase the sensitivity of cleavage-based assays for BoNT detection. Furthermore, the application of the Taguchi DoE approach shows how the method helps to rationally improve enzymatic assays.

## 1. Introduction

The anaerobic spore forming bacteria of the genus *Clostridia* produce botulinum neurotoxins (BoNTs), the most poisonous toxins known and cause of the life-threatening disease botulism [1,2]. BoNTs are synthesized as 150 kDa holotoxins and become activated by cleavage into a 50 kDa light chain (LC) and a 100 kDa heavy chain (HC) which remain connected via a single disulfide bond [3]. The HC can be further divided into a 50 kDa C-terminal (H_C_) and a 50 kDa N-terminal (H_N_) fragment. The LC represents the enzymatic subunit enabling the toxin-mediated cleavage of soluble *N*-ethylmaleimide-sensitive-factor attachment receptor (SNARE) proteins by a Zn^2+^-dependent endopeptidase activity.

The mechanism of BoNT action involves three main steps: First, the binding of the H_C_ to specific receptors on synaptic membranes and subsequent internalization into recycling synaptic vesicles. Second, the LC is translocated by the H_N_ into the cytoplasm. The third step is the LC-catalyzed hydrolysis of the proteins of the SNARE complex, a hydrolysis which prevents the fusion of neurotransmitter-loaded vesicles with synaptic membranes, thereby blocking neurotransmitter release into the synaptic cleft [4,5].

Until now, seven confirmed BoNT serotypes, A–G, have been described [2,4,6]: Serotypes BoNT/A, B, E, and F are mainly pathogenic to humans, whereas BoNT/C and D and their mosaic variants CD and DC cause botulism in animal husbandry and birds [2]. BoNT/G, which was first isolated from Argentinian soil, has not been shown to cause botulism in humans or animals [7,8]. Recently, detailed genetic and proteomic comparisons have revealed that serotypes A, B, E, and F can be divided into more than 40 subtypes based on their amino acid sequences, reactivity to antibodies, and/or functional activity [9,10,11,12].

The different serotypes cleave certain proteins of the SNARE complex at unique sites [13]. BoNT/A, C, and, E cleave the synaptosome-associated protein of 25 kDa (SNAP-25), and BoNT/F, D, and B cleave the vesicle-associated membrane protein (VAMP-1 and -2) [14,15,16,17,18]. BoNT/C is the only serotype that recognizes two substrates by cleaving SNAP-25 and syntaxin [19,20,21,22]. As a common theme, all the subtypes of a given serotype use the same distinct cleavage site on their respective SNARE protein substrate. An exception to this rule is subtype BoNT/F5, which cleavages VAMP-2 four amino acids upstream of all other BoNT/F subtypes [23]. Along this line, a recently identified BoNT molecule originally termed BoNT/H or mosaic BoNT/FA or BoNT/HA has been shown to represent a hybrid molecule composed of BoNT/A H_C_ and a novel LC-H_N_ using the same cleavage site on VAMP-2 as BoNT/F5 [24,25]. Furthermore, advances in bioinformatics and next generation sequencing have allowed for the identification of novel serotypes termed BoNT/X and eBoNT/J (aka BoNT/En) with unique substrate specificities [26,27,28,29]. Though those molecules clearly represent interesting and possible leads for potential novel therapeutic applications of BoNTs, their relevance from a public health point of view as causes of disease is presently unclear. Besides the lack of clinical intoxication by those molecules, they seem to exhibit a rather low intrinsic toxic activity, presumably due to a lack of recognition of suitable entry-mediating receptors and/or overall low or questionable protein expression.

Due to their high potency with an approximately parenteral lethal dose (LD_50_) of 1 ng/kg body weight in humans (reviewed in [1]) and their potential use as a biological weapon, BoNTs are listed as a Tier 1 select agents by the US Center for Disease Control and Prevention (CDC) [30]. The potentially life threatening illness botulism in humans and animals demands highly sensitive BoNT diagnostics [31]. To this end, numerous assays for sensitive BoNT detection have been developed based on immunological, spectrometric, and functional approaches (or combinations thereof) [32]. In all those approaches, the large variability within the BoNT family poses a challenge for detection, since any diagnostic approach has to ensure that no single sero-/subtype escapes detection [32]. Monitoring the substrate cleavage activity for the diagnostics of BoNT has proven to be advantageous for several reasons: (i) The variability of the more than 40 BoNT molecules can be reduced to monitor enzymatic activity at eight cleavage sites; (ii) enzymatic assays are usually more sensitive than pure immunological approaches due to the inherent amplification of signal intensity (one enzyme molecule is able to cleave numerous substrate molecules); (iii) enzymatic assays demonstrate not only the mere presence of BoNTs but also their activity.

In order to monitor the endopeptidase activity of BoNTs, different methods have been used to highlight substrate cleavage in vitro. Förster resonance energy transfer (FRET)-based endopeptidase-assays measure the cleavage of serotype-specific peptide substrates labelled with fluorescent reporter molecules by a change in fluorescent intensity [33,34,35,36,37,38,39,40,41]. Another approach relies on the immunodetection of cleavage products by neo-epitope-specific antibodies that recognize only the cleaved and not the uncleaved substrate [42,43,44,45,46,47,48,49,50,51,52]. Furthermore, cleavage products can be detected by mass spectrometry, where the cleavage of serotype-specific peptide substrates is analyzed by monitoring unique peptide products [53,54]. Since substrate cleavage is monitored in all those assays, assay sensitivity strongly depends on optimal the cleavage conditions for BoNT LCs.

Efficient BoNT substrate cleavage is affected by several factors. Besides cleavage duration, with long incubation times that allow for more complete substrate cleavage, temperature also strongly influences enzyme activity, with an optimal cleavage at 37 °C for most serotypes [55,56] but higher temperatures for others [57]. In addition, the choice of buffer conditions such as pH and NaCl concentration [58] can dramatically affect the catalytic activity of BoNTs (summarized in Appendix A). For the reduction and subsequent release of the LC from H_C_, a reducing agent such as DTT (dithiothreitol) or TCEP (tris(2-carboxyethyl) phosphine) is critical for BoNT’s enzymatic activity in vitro [45,59]. Additionally, since BoNT is a Zn^2+^-dependent metalloprotease, the presence of Zn^2+^ ions is also crucial for substrate cleavage [58]. However, high ZnCl_2_ concentrations can inhibit cleavage when DTT is present due to complex formation with Zn^2+^ ions, as has been shown for LC/A [59]. Hence, buffer compositions have to be carefully balanced for optimal results. Further additives such as BSA (bovine serum albumin) and Tween 20 exhibited stimulating effects on serotypes A and E, but those additives might compromise other LCs [44,45,60,61]. For the osmolyte trimethylamine N-oxide (TMAO), a positive influence on the catalytic activity of LC/A, B, and E has been explained by hypothesizing a stabilizing effect on the molten-globule state [61].

In most previous studies, the influence of only selected factors on the cleavage of the LCs of a few serotypes was analyzed. However, different factors may influence each other. A comprehensive analysis with all BoNT serotypes using multiple factors has not yet been undertaken. To address this issue, we systematically analyzed cleavage conditions for BoNT/A–F. For our investigation, we employed the Taguchi design-of-experiment method (DoE) that makes use of orthogonal arrays to design multifactorial experiments [62]. By systematically and simultaneously varying several factors and then subsequently calculating the impact of single factors by statistical means, the number of required experiments can be drastically reduced by approximately 90%. A statistical analysis of Taguchi experiments includes an analysis of mean (ANOM) to identify optimal factor levels as well as an analysis of variance (ANOVA) to determine each factor’s impact. The DoE method has been broadly employed for process optimization in the engineering and manufacturing industry, whereas it is not widely used in the biomedical field. Still, there have been some reports on the optimization of protein expression or assay development [63,64,65,66]. This method provides a straightforward tool to rationally analyze factors affecting BoNT substrate cleavage with limited experimental efforts. Here, we present a thorough analysis of the factors affecting substrate cleavage of BoNT/A–F using the Taguchi DoE method. We successfully identified DTT, Zn^2+^, and TMAO as critical factors for most BoNT serotypes. In addition, we designed optimal buffer conditions for each serotype individually, thus leading to improved substrate cleavage compared to previously used BoNT cleavage buffers. Finally, we propose a consensus buffer for the enhanced substrate cleavage of all serotypes tested.

## 2. Results

### 2.1. Experimental Design of Taguchi Experiments to Optimize Buffer Composition for BoNT/A–F Substrate Cleavage

The aim of this work was to optimize substrate cleavage of BoNT serotypes A–F by varying buffer composition and frequently used additives. Furthermore, we determined differences in the requirements for optimal cleavage between the different serotypes. Since in a diagnostic setting the serotype present in the suspect sample is unknown, we finally aimed at identifying a consensus buffer that could enable optimized substrate cleavage for all serotypes. Such a comprehensive comparison of cleavage conditions for BoNT/A–F has not yet been undertaken, though it would be highly relevant for public health institutions in charge of botulism diagnostics. To address our aims, we employed a multifactorial Taguchi-DoE approach to reduce the number of required experiments, as well as to elucidate the impact of each factor by statistical means.

The principle workflow of Taguchi experiments is outlined in Figure 1. First, the overall aim of the analysis (here “increase substrate cleavage of BoNT/A–F”) is formulated. Next comes the planning of the individual experiments to be performed. To this end, variable test parameters (factors and levels to be analyzed) as well as all fixed test parameters have to be defined. For our experiments, we applied a short incubation time of 30 min as a fixed parameter, as rapid detection is important for diagnostic purposes. As the only exception, the incubation time was extended to 18 h for BoNT/C due to its low cleavage activity after 30 min. As an additional parameter, the incubation temperature was set to 37 °C, which in a diagnostic setting would be optimal where rapid detection is paramount. A basal 50 mM HEPES (4-(2-Hydroxyethyl) piperazine-1-ethanesulfonic acid) buffer was used to test the variable test parameters, which were chosen according to their known influence on the catalytic activity of BoNT in previous works (Appendix A). In total, we investigated seven different factors (pH, ZnCl_2_, DTT, NaCl, BSA, TMAO, Tween 20), each at three different levels as variable test parameters.

Furthermore, the measured variable, which provides the data for the statistical analysis, has to be defined. We chose to analyze the cleavage rate of full-length recombinant substrates measured densitometrically after SDS-PAGE and Coomassie staining. By employing this setup, we circumvented any additional technical read-outs that might have overshadowed the efficiency of the cleavage reaction itself (e.g., by varying quality or affinity of antibodies used for detection of cleaved or uncleaved substrate, influence of substrate length etc [42,59]).

Finally, the experimental layout, which depends on the number of tested factors and levels, is determined using standardized orthogonal arrays. Theoretically, for analyzing the substrate cleavage of one serotype by testing seven factors in three different levels, an L18-array is applicable [62]. However, to facilitate experimental procedures, we decided to divide our test factors in buffer components (pH, ZnCl_2_, DTT, NaCl) and buffer additives (BSA, TMAO, Tween 20) thus allowing the usage of two consecutive L9-arrays (Table 1). With this approach, the analysis of all six serotypes required 108 experiments, whereas 972 experiments would have been necessary for a full factorial experimental setting. This highlights the dramatic reduction of experiments by the Taguchi-DoE-method.

After the planning phase, all experiments are performed according to an orthogonal array layout. Eventually, data are analyzed by an ANOM and an ANOVA to elucidate the contribution of each factor and level tested. In the ANOM, the optimal level of each factor is determined. Here, for each experiment, a signal-to-noise-ratio (S/N-ratio) was calculated using a specific target function [62]. Since we aimed at maximizing substrate cleavage, we used the larger-the-better function to transform the percentage substrate cleavage of each experiment in S/N-ratios. Then, experiments with the same factor level were grouped, and respective mean S/N-ratios were calculated. Hereby, the optimal factor level was determined by a maximum mean S/N-ratio. The ANOVA compared variances within experiments with the same factor (but at different levels) to identify predominant factors and the role each factor plays on substrate cleavage.

After completing data analysis, final conclusions can be drawn. In our case, this was the optimal buffer composition for each serotype tested as well as the identification of the most relevant factors. Finally, to validate the results of the Taguchi experiment, control experiments, comparing the newly empirically optimized settings to previously used settings, are performed.

### 2.2. Buffer Composition for Optimal Cleavage Varies between Different BoNT Serotypes

To identify optimal cleavage conditions for BoNT serotypes A–F, we performed two consecutive L9-arrays testing (1) buffer components and (2) buffer additives (Table 1). In the first L9-array (1), we investigated the effects of pH, ZnCl_2_, DTT, and NaCl. The tested factors and levels (e.g., pH value or concentrations of the compounds) were probed according to the predetermined orthogonal array layout (Table 1) in nine experiments for each serotype with two independent biological repetitions. As these factors are known to influence the substrate cleavage of different serotypes, we expected to observe differences between the different buffer compositions. Indeed, differences in cleavage efficiency between the different experimental conditions became obvious (Figure 2, left images of each panel). For instance, Experimental Condition 6 (pH 7.0, 250 µM ZnCl_2_, 1 mM DTT, 20 mM NaCl) was detrimental for cleavage by BoNT/A, B, E, and F, whereas substrate cleavage was much more efficient in Experimental Condition 5 (pH 7.0, 50 µM ZnCl_2_, 25 mM DTT, 0 mM NaCl).

Interestingly, experimental conditions supporting the enzymatic activity of BoNT/D and especially BoNT/C, both not inducing natural disease in humans, seemed to differ from conditions for serotypes pathogenic to humans (A, B, E, and F). For example, Buffer 6 had a negative influence on all serotypes except for BoNT/C. Similarly, Buffer 3 only influenced BoNT/D substrate cleavage in a negative way. These results indicate that while the cleavage conditions for serotypes pathogenic to humans are more or less similar, the “veterinary” serotypes C and D require different buffer compositions for optimal cleavage.

With the results of the first L9-array experiments, we determined the optimal levels of respective factors for each tested serotype (see below). Having adjusted optimal levels, we performed the second L9-array with the additives BSA, TMAO, and Tween 20 (Table 1). Here, both BoNT/C and F exhibited similar cleavage in the nine different buffers, thus indicating a low effect of the additives on respective serotypes. Contrary, the cleavage activity of BoNT/A, B, D, and E differed in the tested buffers (Figure 2, right images of each panel). Cleavage was strongly reduced in Buffers 3, 6, and 9, all of which contained 1.5 M TMAO. These results indicate a detrimental effect of high TMAO concentrations on the catalytic efficiency of these serotypes.

### 2.3. ANOM and ANOVA Reveal Optimal Buffer Composition and Impact on Cleavage Efficiency

To find optimal cleavage conditions for each serotype and to quantify the magnitude by which each factor influences cleavage, statistical analysis was carried out. Due to the fractional factorial design of the two orthogonal L9-arrays, the effects of several factors were overlaid. An ANOM allowed for the identification of the optimal level of each factor tested. As cleavage data were transformed using the larger-the-better function, a maximum in the S/N-ratio indicated optimal factor levels.

An analysis of the first L9-array experiments (Figure 3, left images of each panel) revealed that most serotypes prefer a neutral pH between 7 and 7.5, a moderate ZnCl_2_ concentration (50 µM), and a high DTT concentration (25 mM). Exceptions were BoNT/C and, in some instances, BoNT/D. In contrast to all other serotypes, optimal cleavage conditions for BoNT/C could be found at a lower pH of 6.5, a higher ZnCl_2_ concentration (250 µM), and a lower DTT concentration (5 mM). BoNT/D shared an optimal DTT concentration with BoNT/C (5 mM), but, contrary to BoNT/C, cleavage was inhibited by high ZnCl_2_ concentrations (>10 µM). For BoNT/B and E, a slight inhibitory effect of NaCl on cleavage efficiency could be seen. In that sense, the results from the ANOM depicted the trends that became obvious from the matrix experiments: Better cleavage of the serotypes pathogenic to humans at high DTT and moderate ZnCl_2_ levels, as well as distinctively different optima for each BoNT/C and D.

A similar picture was evident after the ANOM analysis of the second L9-array. Here, a strong inhibition at a high TMAO concentration was obvious for BoNT/A, B, D, and E. However, except for D and E serotypes, a slight enhancement of cleavage could be shown at 0.75 M TMAO (Figure 3, right images of each panel). An analysis of additives further revealed that BSA had a slightly inhibitory effect on BoNT/A and F, a stimulatory effect on BoNT/C, and a negligible effect on BoNT/B, D, and E. Tween 20 was a negligible factor for most serotypes, yet it exhibited a stimulatory effect on BoNT/B substrate cleavage. However, except for TMAO, the overall influence of buffer additives on cleavage efficiency was rather small compared to the influence of buffer compositions tested in L9-Array 1.

Finally, the impact of each factor was determined by an ANOVA. These results, combined with the optimal factor levels determined by an ANOM, are summarized in Table 2 (for more detailed results, see Appendix A). Here, a high percentage of factor impact indicates that the specific factor had a large influence on substrate cleavage, either positively or negatively, whereas a low value marks a factor as negligible. As expected, ZnCl_2_ and DTT both had the largest impact on BoNT cleavage regarding buffer composition, while NaCl concentration and pH were of lower importance. However, the optimal levels for each serotype, especially regarding ZnCl_2_, were markedly different. Regarding buffer additives, only TMAO had a significant impact on cleavage, mostly due to the strong inhibition observed at high concentrations. Since only three factors were tested in the second L9-array, the empty factor (not used for testing different levels of a potential fourth factor) was used to control for artefacts due to interactions between different factors. Here, non-significant factor impact levels below 8% indicated that the results were not skewed due to interactions.

Summing up, these results demonstrate that different BoNT serotypes prefer different buffer compositions for optimal substrate cleavage, with BoNT/C differing most from the other serotypes. Furthermore, DTT, TMAO, and ZnCl_2_ were identified as factors with the highest impact on cleavage efficiency.

### 2.4. Optimized Buffers Enhance BoNT Substrate Cleavage Compared to Reference Buffers

Based on the ANOM and the ANOVA, optimized buffers for each individual serotype were determined and compared to two previously described and often cited reference buffers used for BoNT substrate cleavage. These reference buffers were entitled ‘Jones Buffer’ (50 mM HEPES, 10 µM ZnCl_2_, 5 mM DTT, 0.5% Tween 20, 1 mg/mL BSA, pH 7) which had been optimized for an Endopep-ELISA assay for the detection of BoNT/A and BoNT/E [45], and ‘Evans buffer’ (50 mM HEPES, 20 µM ZnCl_2_, 1 mM DTT, 1% BSA, pH 7.4), which was used in an ELISA-based assay for detecting the endopeptidase activity and receptor binding of BoNT/A, B, and F [67].

In the optimized buffers identified in this work (Table 2), substrate cleavage was more efficient compared to the two reference buffers tested for all serotypes (Figure 4 and Table 3). The improvement of cleavage in the optimized buffers was more pronounced when compared to the ‘Evans buffer’ and less prominent when compared to the ‘Jones buffer.’ Except for BoNT/D, which was robust towards changes in the three buffers tested, a two- to 34-fold lower BoNT concentration was sufficient for 50% cleavage in the optimized buffers (Figure 4 and Table 3). The most striking improvement could be achieved for BoNT/C. A 206-fold lower BoNT/C concentration in the optimized buffer was sufficient for 50% cleavage when compared to the ‘Jones buffer.’ Almost no cleavage was observed in the ‘Evans buffer.’

In summary, our data show that despite seemingly minor changes, the choice of adequate buffer conditions can have a tremendous influence on the efficiency of BoNT substrate cleavage. Furthermore, these results show how Taguchi-DoE optimization can help in identifying critical factors and optimal concentrations for enhanced cleavage.

### 2.5. Design of a Consensus Buffer for all Serotypes

For a diagnostic setting when unknown BoNT serotypes have to be identified, a single cleavage buffer is highly desirable. Therefore, we aimed to create a consensus buffer in which all six serotypes sufficiently cleave their substrate. Since each serotype requires unique conditions for optimal cleavage, the design of such a consensus buffer is challenging. Factors with large impact such as DTT, TMAO, and ZnCl_2_ have to be carefully adjusted for each serotype. This is particularly important, since the optimal levels of these factors vary between the different serotypes.

In our consensus buffer, we chose DTT and TMAO concentrations of 25 mM and 0.75 M, respectively, at a pH of 7 because the majority of serotypes showed good catalysis at these levels. In order to support cleavage by BoNT/C, we applied a high ZnCl_2_ concentration (250 µM). NaCl and BSA were omitted due to the partial inhibition of some serotypes, whereas Tween 20 was included at a concentration of 1% mainly due to its positive effect on BoNT/B. For BoNT/D and E, less efficient cleavage was anticipated in the consensus buffer, as both showed inhibition by high ZnCl_2_ concentrations as well as medium TMAO concentrations in the ANOM.

Basically, a consensus buffer has to find a compromise between different enzymatic requirements; therefore, cleavage in the consensus buffer was not as efficient for all serotypes compared to individually optimized buffers (Figure 4 and Table 3). However, the cleavage of BoNT/B, C, and F was still more efficient compared to the reference buffers (Evans and Jones), and the cleavage of BoNT/A was still more efficient compared to the ‘Evans buffer.’ As expected, the cleavage of serotypes D and E was reduced compared to the individually optimized buffers, which showed that accurate predictions of expected efficiencies can be made based on an ANOM and an ANOVA. Still, our newly designed consensus buffer enables the simultaneous detection of enzymatic activities of all relevant BoNT serotypes without the requirement to measure a given sample in various buffer systems.

## 3. Discussion

### 3.1. The Taguchi DoE Method Enables Identification of Optimal Cleavage Conditions for each Serotype

Many diagnostic assays for the detection of botulinum neurotoxins rely on the enzymatic cleavage of their cognate substrate. To achieve maximum sensitivity in the detection of these highly potent toxins, there is a common need for the optimization of catalytic efficiency. As most assays are developed to simultaneously detect single or few different serotypes, no extensive comparison of the requirements regarding buffer composition and additives has been done so far. Optimization following the classical “one-factor-at-a-time” approach offers the advantage of covering all possible combinations, thereby determining optimal conditions independent of mutual interactions between different factors [68]. However, this strategy is time-consuming and tedious and, therefore, not easily applicable for a comprehensive analysis of a large number of test factors and levels. On the contrary, the Taguchi DoE method offers the advantage of testing multiple factors at several levels in a factorial design using pre-existing orthogonal arrays, thereby greatly reducing the number of experiments necessary for assay optimization. Despite its many advantages compared to full factorial experiments, the Taguchi DoE method is still barely used in the life sciences. However, by applying this method, significant improvements can be gained in a shorter amount of time while the statistical analysis dissects influence from negligible factors.

One main criticism of DoE methods is that interactions between different factors tested might remain unrecognized or even falsify the outcome [69]. Due to the fact that not all possible combinations are tested in factorial design experiments, this may hold true. However, due to the large number of necessary experiments, the full factorial testing of all conditions is often not feasible. Hence, the DoE method often delivers a more complete and systematic dataset as compared to the testing subsets of conditions without statistical considerations and analysis. Moreover, in this work, we did not find any major interactions between different factors tested—as proven by the low factor impact of the empty control factor in the second L9-array.

### 3.2. Previously Published Results could be Confirmed by the Taguchi DoE Analysis

With our analysis, we were able to confirm the impact of several factors on BoNT cleavage that have been described previously. For instance, our analysis revealed that NaCl negatively influences the catalytic activity of BoNT/B and BoNT/D. This is in agreement with the results of Shone et al., who observed that increasing NaCl concentrations had an inhibitory effect on BoNT/B substrate cleavage for unknown physiological reasons [58]. Similar to our results, Jones et al. demonstrated that Tween 20 enhances the substrate cleavage of BoNT/A and E [45]. Furthermore, we found a highly increased cleavage for BoNT/C at pH 6.5 and high ZnCl_2_ concentrations, which is supported by the observations of Moura et al., who also found that a pH of 6.5 was optimal for BoNT/C cleavage [57]. Regarding DTT, concentrations of 25 mM were shown to increase the activity of BoNT/A, B, E, and F in our work. Likewise, Wang et al. demonstrated an enhanced substrate cleavage of BoNT/B at 20 mM DTT [70]. Mizanur et al. also observed an increased LC/A activity with increasing DTT concentrations, although they found a slight inhibition at concentrations above 4 nM [59]. Additionally, a strong inhibition of the substrate cleavage of LC/A at concentrations above 100 µM ZnCl_2_ was observed in the same work, which is also in agreement with our results.

We did, however, also obtain deviant results for some factors and concentrations experimentally tested. For BoNT/A and E, optimal cleavage was observed at concentrations of 5 and 2.5 mM DTT, respectively, by Shone et al. [58], which is lower than the optimal concentration of 25 mM in our work. It is likely that these disagreements can be explained by differences in the experimental setup and cleavage time. In our assay, we used a very short incubation time of 30 min. This might have required a much higher DTT concentration for the sufficient reduction of BoNT/A and E. On the other hand, longer incubation times like those used by Shone et al. might have allowed for lower DTT concentrations for sufficient reduction. Likewise, BSA had little impact on cleavage in our setting. However, the addition of BSA to minimize adsorption to microwells at low toxin concentrations might increase cleavage efficiency in plate-based assays with detection by cleavage-specific antibodies [43,44,45].

In our work, we observed a strong inhibition of cleavage at TMAO concentrations above 0.75 M. On the contrary, Nuss et al. found a strong increase of LC/A, B and E activity with increasing TMAO concentrations up to 2 M [61]. Unlike Nuss et al., who investigated cleavage by isolated toxin light chains, we tested full-length non-reduced toxins. The osmolyte TMAO is thought to have a positive effect on protein structure by enhancing a more compact structure [71,72]. This might explain the enhanced cleavage of the isolated light chain while the full-length toxin was inhibited by sterically hindering the accessibility of the reducing agent and the release of the active light chain.

## 4. Conclusions

In this work, we presented, for the first time, a rational analysis of the factors influencing substrate cleavage, covering all clinically relevant BoNT serotypes (except G, FA, and F5). Using the Taguchi DoE method, we identified critical factors for each serotype and designed optimal buffer conditions that led to improved substrate cleavage. Additionally, we proposed a consensus buffer recipe in which all serotypes exhibited relatively good cleavage and which could be applied to all serotypes if the serotype is a priori unknown. This is of particular importance in a diagnostic setting, where the serotype is not known prior to analysis. Hereby, our optimized cleavage buffers are instrumental to the improvement of cleavage based assays for BoNT detection. In our recent work, the consensus buffer was successfully implemented in a neoepitope-based suspension array for the functional detection of botulinum neurotoxin serotypes A–F [52]. Finally, we demonstrated that the Taguchi DoE method is a straightforward tool that is transferable to optimize other multi-parameter enzymatic reactions.

## 5. Materials and Methods 

### 5.1. Chemicals and Toxins

For the optimization of BoNT cleavage, the following reagents were used: Zinc chloride (Merck, Darmstadt, Germany), bovine serum albumin (BSA; Roth, Karlsruhe, Germany), Tween 20 (Merck, Darmstadt, Germany), trimethylamine N-oxide dihydrate (TMAO; Sigma-Aldrich, Taufkirchen, Germany), dithiothreitol (DTT; Roth, Karlsuhe, Germany), NaCl (neoLab Migge, Heidelberg, Germany). Purified botulinum neurotoxin serotypes A, B, C, D, E, and F were all obtained from Metabiologics (Madison, WI, USA). It was previously shown that BoNT/D from Metabiologics is actually the mosaic variant BoNT/DC [57]. However, this work only depicted the light chain activity, and the light chains of BoNT/D and BoNT/DC are almost identical [73]. Therefore, following the manufacturer, the designation used for the toxin in this manuscript was BoNT/D. Toxins were handled in a biosafety cabinet in a dedicated toxin laboratory and decontaminated with 5% NaOH for 24 h.

### 5.2. Expression and Purification of Full-Length SNAP-25 and VAMP-2

Recombinant rat SNAP-25 amino acid 1-206 fused to a C-terminal His6tag (rSNAP-25H6) was expressed in an *Escherichia coli* M15 strain, as described previously [16]. The plasmid pET15b-VAMP-2-encoding rat VAMP-2 amino acid 1-97 that was fused to an N-terminal thrombin cleavable His6tag (H6trVAMP-2 1-97) was expressed in an *E. coli* BL21DE3 strain upon induction by IPTG61. *E. coli* cells were harvested, resuspended in 50 mM Tris-HCl, pH 8.0, 150 mM NaCl, 5 mM imidazole and protease inhibitor EDTA-free complete (Roche, Mannheim, Germany) and lysed by ultrasound. rSNAP-25H6 and H6trVAMP-2 1-97 were isolated by immobilized metal affinity chromatography using a Talon matrix (Takara Bio, Mountain View, CA, USA), washed in resuspension buffer supplemented with 1 M NaCl, and eluted in a resuspension buffer supplemented with 250 mM imidazole. The proteins were polished by gel filtration (Superdex-75, GE Healthcare), rSNAP-25H6 in 20 mM HEPES-KOH, pH 7.4, 150 mM KCl, and H6trVAMP-2 1-97 in phosphate buffered saline (PBS), pH 7.4. Fractions containing the recombinant proteins were pooled, frozen in liquid nitrogen, and kept at −70 °C.

Protein concentrations were determined subsequent to 15% SDS-PAGE and Coomassie blue staining by using a LAS-3000 imaging system (FUJIFILM Europe GmbH, Düsseldorf, Germany), the AIDA 3.51 software (Raytest, Straubenhardt, Germany), and BSA (100–1600 ng) as reference protein.

### 5.3. Design of Taguchi Experiments 

The Taguchi-DoE method generally aims at optimizing processes. It can be used to determine the influence of several factors of interest tested at different levels to optimize a specific process. The experimental layout is adopted from predefined orthogonal arrays that best suit the necessary number of factors and levels to be tested [74]. Available arrays range from two factors tested at two levels each (L4-array) to 31 factors tested at up to 5 levels. In this work, we aimed at optimizing BoNT substrate cleavage by varying buffer composition and frequently used additives. Each factor was tested at three levels in a range of concentrations that had a significant influence on BoNT cleavage in previous works (Appendix A). Based on the number of factors (7) and levels (3 for each parameter), an L18-array would have been theoretically suitable to simultaneously test all factors and levels. However, we decided to use two consecutive L9-arrays instead (Table 1). This way, the influence of buffer composition and additives could be independently analyzed with the same number of experiments required. In the first L9-array, the buffer pH and different concentrations of ZnCl_2_, DTT, and NaCl were analyzed. Subsequently, optimal conditions were adjusted, and, in the second L9-array, the frequently used additives BSA, TMAO, and Tween 20 were tested. All buffers were based on 50 mM HEPES, and each condition was tested in two independent biological experiments.

### 5.4. Analysis of Substrate Cleavage by SDS-PAGE

Cleavage reactions were carried out in a PCR-thermocycler (Eppendorf, Hamburg, Germany) in a volume of 30 µL of the assigned buffer for 30 min (BoNT/A, B, D, E and F) or 18 h (BoNT/C only), respectively, at 37 °C. Toxin concentrations were titrated to achieve substrate cleavage between 20% and 60% in the basal HEPES-buffer with 5 mM DTT added (first L9-array), or in the optimized buffer without further additives (second L9-array) to enable the monitoring of enhancing or inhibitory effects.

In the first L9-array experiments, 3 µM SNAP-25 was incubated with 0.3 nM BoNT/A, 50 nM BoNT/C, or 25 nM BoNT/E (non-trypsinized), while 20 µM VAMP-2 was incubated with 100 nM BoNT/B, 80 nM BoNT/D or 5 nM BoNT/F. As experiments of the second L9-array were carried out with optimized cleavage buffers resulting from the first L9-array, toxin concentrations were reduced to 50, 10, 40 and 0.625 nM for BoNT/B, C, D, and F, respectively.

After digesting, samples were heated for 10 min at 95 °C to inactivate the toxin, and 15 µL of 3 × Laemmli loading buffers (150 mM Tris/HCl pH 6.8, 6% SDS, 30% glycerol, 7.5% β-mercaptoethanol, 0.25% bromophenol blue) were added. Samples were then reheated at 70 °C for 10 min, and 10 µL were loaded on a 16.5% polyacrylamide-peptide gel for Tricine-SDS-PAGE [75]. After electrophoretic separation, gels were stained with colloidal Coomassie [76] and documented at a ChemiDoc workstation (Bio-Rad, Munich, Germany). The band intensity of the uncleaved substrate and the N-terminal cleavage product was densitometrically determined using Image Lab 5.2.1 software (Bio-Rad, Munich, Germany). Background intensity was subtracted, and the percentage of cleavage y (0 = Uncleaved; 100 = Fully cleaved) was determined by normalizing the N-terminal cleavage product with regard to the uncleaved negative control (in L9-array experiments) or to the sum of uncleaved and cleaved band (control experiments).

### 5.5. Statistical Analysis of Taguchi Experiments

Taguchi experiments were analyzed as described before [62]. Figure 1 gives an overview of the successive steps of a Taguchi experiment. To elucidate the influence of each factor tested on substrate cleavage by BoNT, an analysis of means (ANOM) was performed. To this aim, the determined normalized cleavage y for each experiment (j = 1–9) was transformed to the signal-to-noise-(S/N)-ratio (*η)* using a “the-larger-the-better” log-transformation, where i is the count of each independent experiment (*n* = 2):(1)$$\eta_{j}=-10{\;\log}_{10}\;\left(\frac{1}{n}\;\overset{n}{\underset{i=1}{\sum }}\frac{1}{y_{i}{}^{2}}\right)$$

Next, the mean S/N-ratio (*η*_Factor Level_) for experiments with identical factor levels (e.g., the same pH or the same DTT concentration tested) was calculated as the arithmetic mean. For example, the mean S/N-ratio for each experiment tested with 1 mM DTT was calculated as the arithmetic mean from S/N-ratios from experiments 1, 6, and 8 (see Table 1):(2)$${\bar{\eta }}_{{\mathrm{DTT}}_{1\mathrm{mM}}}=\frac{1}{3}\;\left(\eta_{\mathrm{Exp}.1}+\eta_{\mathrm{Exp}.6}+\eta_{\mathrm{Exp}.8}\right)$$

Trends in cleavage efficiency and optimal factor levels were visualized by plotting respective mean S/N-ratios over the tested factor levels.

To further analyze if the observed differences were statistically significant, an analysis of variance (ANOVA) was performed. To this aim, the mean sum squares (SQ_m_)
(3)$${\mathrm{SQ}}_{m}=\frac{{\left(\sum \eta_{i}\right)}^{2}}{9}$$
and the total error sum square (SQ_total_)
(4)$${\mathrm{SQ}}_{\mathrm{total}}=\;\sum {\eta_{i}}^{2}\;-\;{\mathrm{SQ}}_{m}$$
of all experiments were calculated for each L9-array. From those, the sum square variation (SQ_Factor_)of each factor was calculated:(5)$${\mathrm{SQ}}_{\mathrm{Factor}}=\;\frac{{\left(\sum \eta_{{\mathrm{Factor}}_{\mathrm{Level}\;1}}\right)}^{2}}{3}\;+\;\frac{{\left(\sum \eta_{{\mathrm{Factor}}_{\mathrm{Level}\;2}}\right)}^{2}}{3}\;+\;\frac{{\left(\sum \eta_{{\mathrm{Factor}}_{\mathrm{Level}\;3}}\right)}^{2}}{3}-\;{\mathrm{SQ}}_{m}$$

Then, the impact (*p*) of each factor on substrate cleavage was calculated:(6)$$p\;{\left[\%\right]}_{\mathrm{Factor}}=\;\frac{{\mathrm{SQ}}_{\mathrm{Factor}}}{{\mathrm{SQ}}_{\mathrm{total}}}$$

Finally, the F-value for each factor was calculated by dividing the variance (V_Factor_) of each factor
(7)$$V_{\mathrm{Factor}}=\;\frac{{\mathrm{SQ}}_{\mathrm{Factor}}}{\mathrm{DF}}$$
by the estimation of the variation error (V_F2_) for each L9-array, which was calculated using the two factors with the lowest sum square variation
(8)$$V_{F2}=\;\frac{{\mathrm{SQ}}_{\mathrm{low}\;1+\mathrm{low}\;2}}{\mathrm{DF}}$$
with DF correlating to the degrees of freedom (2 for V_Factor_ and 4 for V_F2_). The determination of the F-value allowed for the calculation of the *p*-value of each factor using the F-distribution.

### 5.6. Validation of Optimized Buffer Performance

Finally, to test if the cleavage efficiency for BoNT serotypes A–F was enhanced after both optimization processes, substrate cleavage was tested against different buffers. Therefore, the dilution series of BoNTs were incubated with SNAP-25 or VAMP-2 in the respective optimized buffers (Table 2). For comparison, two previously described buffers containing either 50 mM HEPES, 20 µM ZnCl_2_, 5 mM DTT, 1 mg/mL BSA, 0.5% Tween 20 at pH 7 (termed the Jones buffer [45]) or 50 mM HEPES, 20 µM ZnCl_2_, 1 mM DTT, 1 % BSA at pH 7 (termed the Evans buffer [67]) were tested in parallel. The amount of cleavage was determined as described above, and values were plotted vs. increasing BoNT concentrations using Michaelis–Menten kinetics, assuming a Km > 0 and Vmax of 100.
(9)$$Y=V_{\max}*\;\frac{X}{\left(K_{m}+X\right)}.$$

## Figures and Tables

**Figure 1 toxins-11-00588-f001:**
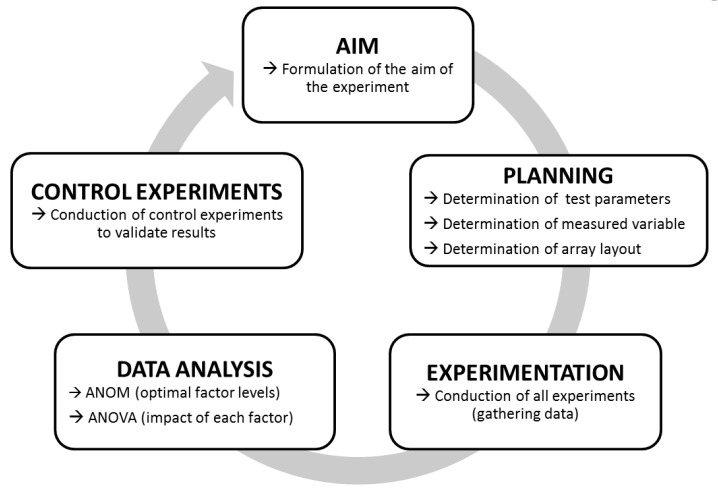
Work-flow of Taguchi experiments. After formulating a specific aim (here, e.g., “Increase the cleavage of synaptosome-associated protein of 25 kDa (SNAP-25) by botulinum neurotoxin/A (BoNT/A)”), fixed and variable test parameters, the measuring method, and an adequate array layout are determined. If all parameters have been determined, experiments are carried out. Finally, the data are analyzed by statistical means using an analysis of mean (ANOM) (to determine the optimal factor levels) and an ANOVA (to determine the impact of each factor). Eventually, the outcome of Taguchi experiments should be controlled to validate the results.

**Figure 2 toxins-11-00588-f002:**
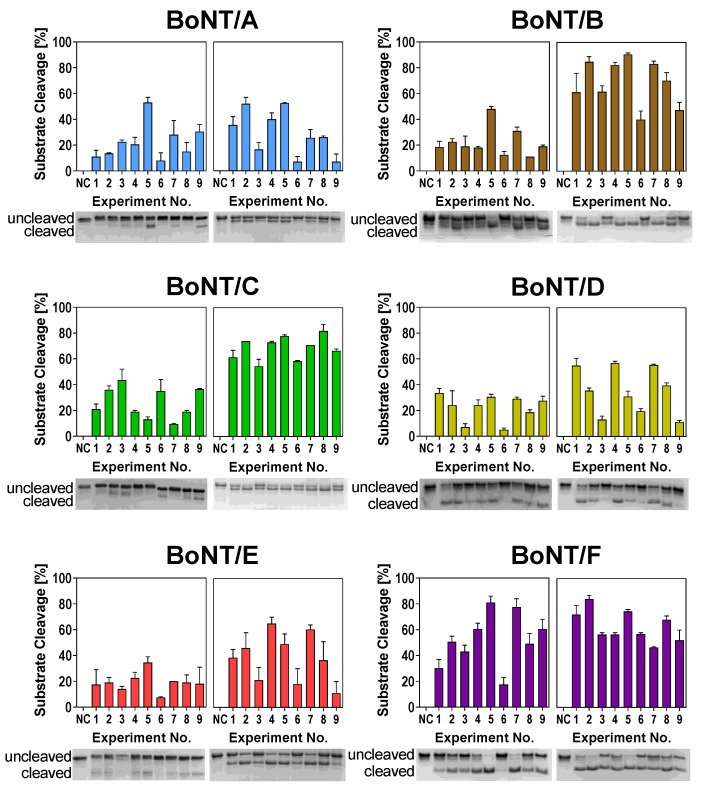
Substrate cleavage of BoNT serotypes A–F in different buffer compositions according to the L9-Array 1 (left image of each panel) and L9-Array 2 (right image of each panel). Full-length SNAP-25 or vesicle-associated membrane protein-2 (VAMP-2), respectively, were incubated with BoNT for 30 min (BoNT/A, B, D, E, F) or 18 h (BoNT/C) at 37 °C in the assigned buffer (Table 1). Cleavage was analyzed densitometrically via SDS-PAGE and Coomassie staining. One representative gel image out of two experiments is shown. Bars indicate normalized averaged cleavage (*n* = 2; standard deviation (SD) is shown). NC = Uncleaved negative control in 50 mM HEPES.

**Figure 3 toxins-11-00588-f003:**
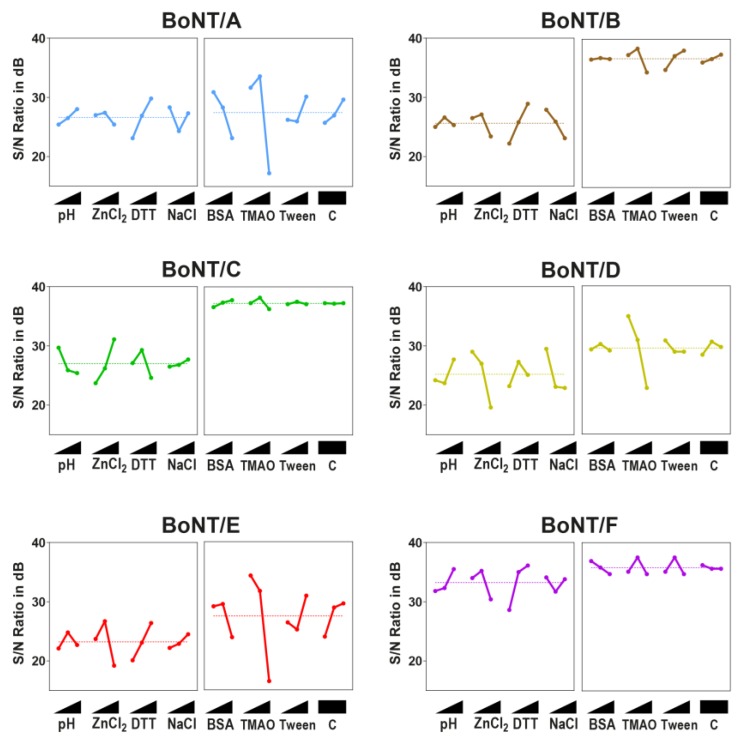
ANOM of L9-Array 1 (left image of each panel) and L9-Array 2 (right image of each panel). signal-to-noise (S/N)-ratios (larger-the-better) in db (decibel) for each experiment were determined. Graphs display the mean S/N-ratio for each factor level and the overall mean out of two independent experiments. Factor levels (from lowest to highest level as indicated with triangle): pH: 6.5, 7, 7.5; ZnCl_2_: 10, 50, and 250 µM; DTT (dithiothreitol): 1, 5, and 25 mM; NaCl: 0 mM, 20, and 100 mM; BSA (bovine serum albumin): 0, 0.5, and 1 mg/mL, trimethylamine N-oxide (TMAO): 0, 0.75, and 1.5 M; Tween 20: 0%, 0.5%, 1%; C = Control (empty factor). A maximum in each factor indicates the optimal level.

**Figure 4 toxins-11-00588-f004:**
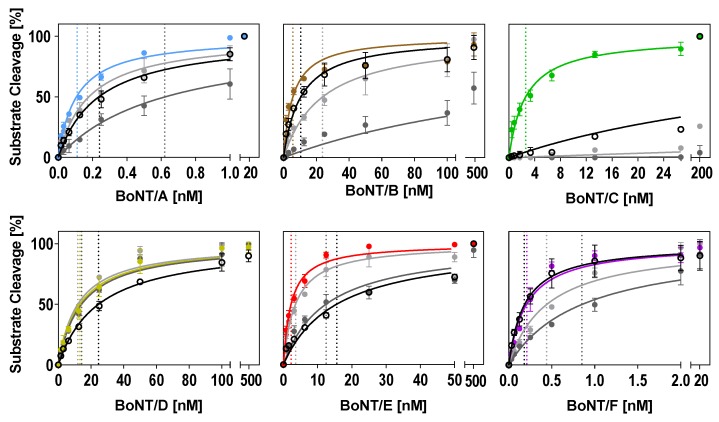
Comparison of optimized buffer conditions for substrate cleavage by each serotype with two reference buffers and our consensus buffer. Either full-length SNAP-25 or VAMP-2 were incubated in a 1:2 dilution series of BoNT for 30 min (BoNT/A, B, D, E, F) or 18 h (BoNT/C) at 37 °C in the serotype-optimized buffer developed in this work (colored circles; see Table 2 for buffer compositions), the Jones buffer (light grey circles), the Evans buffer (dark grey circles), or the consensus buffer (open black circles). *K_m_* values (indicated as dashed lines for each curve) were calculated using Michaelis–Menten kinetics (Y = Vmax × X/(K_m_ + X)) assuming a *K_m_* > 0 and *V_max_* equal to 100. Cleavage was analyzed via SDS-PAGE (*n* = 2; ±SD), as described in the method section.

**Table 1 toxins-11-00588-t001:** Layout of the two consecutive orthogonal L9-arrays. All buffer were based on 50 mM HEPES.

Exp. No.	L9-Array 1: Buffer Composition				L9-Array 2: Buffer Additives		
	pH	ZnCl_2_ [µM]	DTT [mM]	NaCl [mM]	BSA [mg/mL]	TMAO [M]	Tween 20 [%]
1	6.5	10	1	0	0	0	0
2	6.5	50	5	20	0	0.75	0.5
3	6.5	250	25	100	0	1.5	1.0
4	7.0	10	5	100	0.2	0	0.5
5	7.0	50	25	0	0.2	0.75	1.0
6	7.0	250	1	20	0.2	1.5	0
7	7.5	10	25	20	1.0	0	1.0
8	7.5	50	1	100	1.0	0.75	0
9	7.5	250	5	0	1.0	1.5	0.5

**Table 2 toxins-11-00588-t002:** Optimized buffer conditions and factor impact (in %) for each BoNT serotype analyzed according to results of an ANOM and an ANOVA. Numbers in parentheses indicate factor impact.

	BoNT/A	BoNT/B	BoNT/C	BoNT/D	BoNT/E	BoNT/F
pH	7.5 (10%)	7 (3%)	6.5 (22%)	7.5 (10%)	7 (7%)	7.5 (14%)
ZnCl_2_ [µM] ^a^	50 (6%)	50 (18%)	**250 (56%)**	**10 (51%)**	**50 (52%)**	50 (21%)
DTT [mM] ^a^	**25 (61%)**	**25 (52%)**	**5 (21%)**	5 (9%)	25 (36%)	**25 (58%)**
NaCl [mM]	0 (23%)	0 (27%)	*100 (2%)* ^b^	0 (30%)	*100 (5%)* ^b^	0 (7%)
BSA [mg/mL]	0 (15%)	0.2 (0%)	1 (25%)	0.2 (1%)	0.2 (8%)	0 (32%)
TMAO [M] ^a^	**0.75 (76%)**	**0.75 (57%)**	**0.75 (70%)**	**0 (93%)**	**0 (77%)**	**0.75 (60%)**
Tween 20 [%]	1 (5%)	1 (37%)	0.5 (5%)	0 (3%)	1 (7%)	0 (6%)
Control ^c^	n.a. (4%)	n.a. (6%)	n.a. (0%)	n.a. (3%)	n.a. (8%)	n.a. (3%)

^a^ Factors with a statistically significant contribution (*p* < 0.05) are depicted in bold. ^b^ Due to the low impact of NaCl on the cleavage of BoNT/C and BoNT/E, no NaCl was added to the optimized buffer despite a slightly enhanced cleavage by the ANOM (depicted in italics). ^c^ n.a. not applicable.

**Table 3 toxins-11-00588-t003:** Comparison of cleavage efficiency for different buffers (determined with the graphs of Figure 4).

	Toxin Concentration for 50% Cleavage [nM] ^a^			
Serotype	Evans Buffer	Jones Buffer	Optimized Buffer	Consensus Buffer
BoNT/A	0.62 ± 0.05	0.18 ± 0.01	0.11 ± 0.01	0.24 ± 0.01
BoNT/B	193 ± 31.6	23.8 ± 2.29	5.7 ± 0.77	10.5 ± 1.18
BoNT/C	4717 ± 1202	537 ± 37.4	2.6 ± 0.2	53 ± 9.3
BoNT/D	14.2 ± 1.0	11.9 ± 1.2	13.2 ± 1.1	24.6 ± 1.3
BoNT/E	12.5 ± 1.2	3.6 ± 0.26	2.25 ± 0.15	15.6 ± 1.1
BoNT/F	0.85 ± 0.08	0.44 ± 0,05	0.21 ± 0.02	0.18 ± 0.02

^a^ Mean ± SEM from *n* = 2 independent experiments.

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
