# Peer review of "Optimization of SNAP-25 and VAMP-2 Cleavage by Botulinum Neurotoxin Serotypes A–F Employing Taguchi Design-of-Experiments"

_toxins, 2019, doi:10.3390/toxins11100588_

Round 1

Reviewer 1 Report

This article reads very well. The methods and results appear to be well described.

The introduction and Discussion are appropriate lengths and concise enough to understand the study.

I have only one thought for the authors.

I see pH is being a factor for enzymatic activity but could pH not also alter the associated binding of the SNARE proteins and thus maybe produce varying results for the  BoNTs? I am not aware of research into the effects of pH on the binding properties of the SNARE - SNAP proteins.

Author Response

Answer: Thanks for this interesting input. In vivo, the SNARE complex forms in the cytosol at neutral pH out of SNAP-25, syntaxin and VAMP. Possibly this formation might be influenced by a change in pH, but we are unaware of any work analyzing this issue.

Apart from this, the issue does not apply to our work as we use the individual SNARE components SNAP-25 or VAMP in our assay, thus SNARE complex formation is not part of the assay. Similar settings are used in other diagnostic assays which measure the endopeptidase activity in vitro, e.g. by Endopep-MS assay.

Reviewer 2 Report

At line 32: NSF to state full letter, not in the acronyms.

This study would be a novel approach to form a single consensus buffer for detecting all active botulinum neurotoxins.

Your attempt to analyse factors influencing substrate cleavage covering all clinically relevant BoNT serotypes would be valuable, but your research does not suggest that you can find all botulinum toxin activation in a single consensus buffer solution.

I look forward to producing better results based on the results of this experiment.

Reviewer 3 Report

The manuscript clearly defines the study parameters and the application of the Taguchi DoE design and analysis. The results are well explained and the graphics serve as good corollaries ot the text. However, I am not sure that an engineering matrix can be applied to a complex biological system in a fashion that produces useful data. The method of detection, using SDS-PAGE and densitometry, does not seem to be the most sensitive or re-producible method for detection. It would be preferable to see such results applied a more rapid and sensitive analysis such as those described in their introduction; but the composite buffer may not be compatible for such assays. Furthermore, the standardization buffer seems to not be favorable for some types, and increases cleavage for others. I understand that this is being done to try and develop a diagnostic assay but I don't feel the methodology in general would support this application. If a diagnostic assay is being performed in the event of a human intoxication, there are more specific methods available. The paper does offer some useful information about buffer composition but I think it falls short of qualifying this method as a valid diagnostic tool. 
